# Enabling PSO-Secure Synthetic Data Sharing Using Diversity-Aware Diffusion Models

No Author Given

No Institute Given

**Abstract.** Synthetic data has recently reached a level of visual fidelity that makes it nearly indistinguishable from real data, offering great promise for privacy-preserving data sharing in medical imaging. However, fully synthetic datasets still suffer from significant limitations: First and foremost, the legal aspect of sharing synthetic data is often neglected and data regulations, such as the GDPR, are largley ignored. Secondly, synthetic models fall short of matching the performance of real data, even for in-domain downstream applications. Recent methods for image generation have focused on maximising image diversity instead of fidelity solely to improve the mode coverage and therefore the downstream performance of synthetic data. In this work, we shift perspective and highlight how maximizing diversity can also be interpreted as protecting natural persons from being singled out, which leads to predicate singling-out (PSO) secure synthetic datasets. Specifically, we propose a generalisable framework for training diffusion models on personal data which leads to unpersonal synthetic datasets achieving performance within one percentage point of real-data models while significantly outperforming state-of-the-art methods that do not ensure privacy. Our code is available at https://anonymous.4open.science/r/Trichotomy-C02B.

**Keywords:** Privacy, Data-sharing, Conditional Image Generation, Diffusion Models

## 1 Introduction

Generative models have recently gained significant attention for their ability to produce highly realistic images, sometimes even deceiving trained clinicians [34]. This opens the possibility of generating synthetic datasets that can be openly shared without the legal constraints of real medical data [33]. However, this potential raises important legal and practical questions. Primarily, these concern (1) the privacy of patients whose data were used during training and (2) the performance of synthetic datasets on downstream tasks.

To address privacy, we consider the General Data Protection Regulation (GDPR), one of the most comprehensive legal frameworks for data protection worldwide. Its central goal is "to protect the fundamental rights and freedoms of natural persons, and in particular their right to privacy, with regard to the processing of personal data" [2]. Importantly, the GDPR applies only if the data

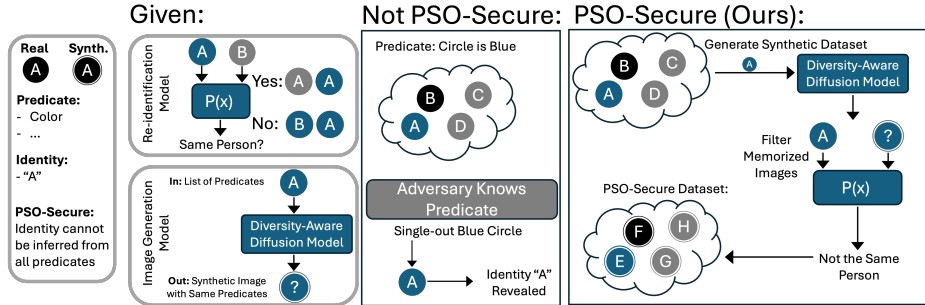

**Fig. 1.** Illustration of PSO-secure dataset generation. A diffusion model, guided by pre-computed features, generates synthetic images that preserve specific predicates (*e.g.*, color) while altering identity. A re-identification model ensures that identities are not retained. As a result, the synthetic dataset maintains relevant attributes of the real data without revealing personal information, making it non-personal under GDPR and suitable for sharing.

in question are considered personal data. The regulation intentionally leaves key terms, including personal, broadly defined to allow case-by-case interpretation. This flexibility enables courts and regulators to balance individual privacy against legitimate public interests [1].

Recital 26 of the GDPR specifies that "the principles of data protection should not apply to anonymous information", meaning data that do not relate to an identified or identifiable person [3]. Therefore, a central legal question becomes whether a given dataset can be considered anonymous. Recital 26 further clarifies that identifiability should be assessed in terms of all means reasonably likely to be used" for identification, including the possibility of singling out an individual [3]. This notion of singling out refers to the use of a combination of observable predicates to uniquely identify a person. Such predicates might include attributes like gender, medical conditions, height, or more context-specific features, for example, the presence of a ring in a radiograph [8].

To formalize this legal concept, Cohen et al. [6] propose a mathematical definition of singling out. They introduce the notion of Predicate Singling-Out security (PSO-security) to describe datasets where such identification is impossible. This framework defines a practical threshold for when synthetic data can be considered legally anonymous under the GDPR. However, identifying all relevant predicates a priori is infeasible in practice, making PSO-secure generation a technically and legally challenging problem.

Even if privacy were fully assured, another fundamental issue remains: the performance of synthetic data. Despite recent advances, generative models have not yet been shown to match real data in downstream tasks. Their primary applications remain in augmentation, imputation, and balancing of real datasets [12,37,40,27,29,7,5,39,26,41,36,34]. If synthetic images were truly equivalent to

real ones, we would not require real data for model training. However, attempts to fully replace real datasets with synthetic counterparts have consistently resulted in performance degradation [9,33,28,31,19,14,17,13,10].

Recent research has therefore focused on evaluating synthetic data quality using metrics beyond image fidelity. Some studies use re-identification models to evaluate temporal consistency in videos [10]. Others emphasize sample diversity as a core metric for synthetic dataset quality [17,10,11]. Despite its relevance, diversity remains under-optimized in current generative approaches.

**Contribution:** We present a framework for generating synthetic datasets that are both PSO-secure and competitive with real data in downstream tasks. Our approach extracts learned predicates from training data and conditions generation on these predicates. By explicitly ensuring that the generated images do not preserve the identity of training samples, we produce synthetic counterparts that share predicates but differ in identity. As a result, singling out is no longer possible, rendering the synthetic data PSO-secure under the GDPR framework. We empirically demonstrate that our method outperforms the current state-of-the-art in downstream performance. Furthermore, we show that models trained exclusively on our synthetic data generalize better than those trained on real data alone. The full pipeline is illustrated in Figure 1.

## 2 Related Work

**Privacy-Preserving Techniques:** Privacy preservation remains a critical challenge in image generation and has been addressed in several studies. Although generative models are designed to avoid the direct replication of real samples, recent work has shown that diffusion models may memorize and inadvertently leak private information if not trained with care [33]. This raises serious concerns for medical data sharing, where regulations such as HIPAA and GDPR mandate strict privacy safeguards. A potential mitigation strategy involves using re-identification models trained to determine whether two samples originate from the same individual [30]. Such models leverage subject labels in the training data to perform re-identification, as demonstrated in [33,10]. While these methods address the technical aspect of privacy and incorporate filtering mechanisms to enforce it, they do not account for the associated legal considerations.

**Image Generation:** Diffusion models have become the leading approach for image generation following their reintroduction with improved noise schedules and architectural enhancements [15]. A major breakthrough was the development of latent diffusion models, which significantly reduced training and sampling times, enabling large-scale commercialization [35]. Subsequent work further optimized schedules and architectures to improve training stability [23,25]. The guidance network is an auxiliary denoising model and has been shown to play a key role in tuning and enhancing generation quality [16]. State-of-the-art methods now employ a less-trained version of the same model to balance efficiency and performance [24]. Despite these advances, class-conditional diffusion models often suffer from limited diversity [11]. To address this, [11] introduced DiADM, a

diversity-aware diffusion model guided by precomputed pseudo-conditional features from a pre-trained network. Using an Inception network as a feature extractor, DiADM separates image quality from diversity, aiming to generate realistic yet varied datasets. However, their evaluation does not consider downstream task performance or privacy implications.

## 3    Methods

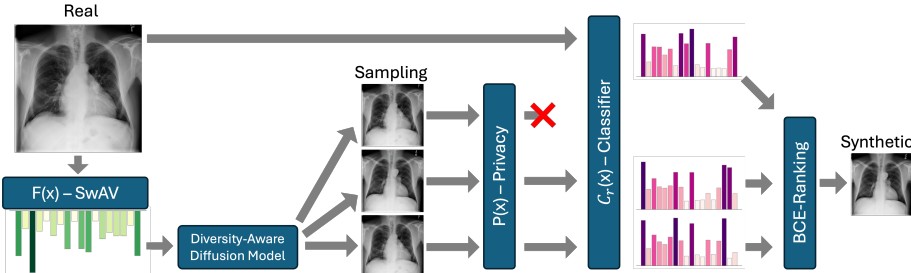

**Fig. 2.** Method illustration: We adapt recently established diversity aware diffusion models (DiADM) [11] to generate multiple images that share the same predicates according to the SwAV feature encoder. A privacy filter ensures that the identity is not preserved. Finally, the images are ranked according to how well they preserve the predicates.

Formally, we aim to generate a synthetic dataset $\mathcal{D}'$ of the same size and the same label distribution as the real dataset $\mathcal{D}$ that achieves comparable performance on a downstream task of predicting $c_d$, while at the same time staying PSO-secure. The labels shall only be used for downstream evaluation, but not for conditioning, to remain generalizable to cases where no data is present. To compare real and synthetic performance, we assume that sharing classifiers is always possible, regardless of their train set. Classification models trained in real data are called $\mathcal{C}_r(x)$, and those trained in synthetic data are called $\mathcal{C}_s(x)$. In practice, hospitals often rely on locally trained models, limiting their ability to benefit from external datasets. Our approach mimics the case, where PSO-secure datasets can be shared across instituations. Therefore, we train a model $\mathcal{C}_s(x)$ on the combined synthetic datasets to demonstrate the potential of using synthetic datasets for the purpose of data sharing. To ensure privacy, we use a re-identification filter which removes privacy violating samples following [10,33]. **Image Generation:** For image generation, we introduce a novel sampling strategy, unique to DiADM-based models. DiADM leverages a knowledge database to improve generative performance by conditioning on pre-trained feature vectors. Inspired by this concept, we adopt a similar approach and observe that the

method proposed by [11] applies the same principle to enhance sample diversity through pseudo-conditional features, denoted as $c_s$. Unlike [11], we use features extracted by a pre-trained SwAV model [4], as we believe these visual features better align with the reconstruction loss of the diffusion model.

Our approach allows us to take the pseudo-label $c_s$ of a real image $x$ which was used to generate the corresponding synthetic image $x'$ and assess memorization by evaluating the prediction of $\mathcal{P}(x, x')$. For each $x \in \mathcal{D}$, we extract $c_s$ and generate a batch of synthetic images $\mathcal{D}'_x$ of size $b = 32$. We then apply a privacy filter to identify and remove any synthetic images that exhibit excessive similarity to real training samples, ensuring that memorization is mitigated. Finally, we select the most suitable synthetic sample by computing the alignment between the real and generated images. This alignment ensures that the predicates of the real image are equal to the predicates of the synthetic image. Specifically, we choose the synthetic image that minimizes the binary cross-entropy (BCE) loss between the predictions of $\mathcal{C}_r(x)$ on the real image $x$ and the synthetic candidate $x'$. Formally, this selection process is defined as:

$$x' = \arg\min_{x' \in \mathcal{D}^-_x} \text{BCE}(\mathcal{C}_r(x), \mathcal{C}_r(x')) \quad \text{for} \quad \mathcal{D}^-_x := \{x' | \mathcal{P}(x, x') = 0\} \qquad (1)$$

where $\mathcal{P}(x, x') = 0$ ensures that knowing predicates $c_s$ does not imply knowledge of the identity. In rare cases where all generated samples are flagged as privacy risks, we reduce the classifier guidance strength by 0.1 and re-sample. The entire sampling process is also visualised in Fig. 2.

## 4   Experiments

**Dataset:** We use MIMIC-CXR (CXR) [22], CheXpert (CXP) [20], and ChestX-ray8 (NIH) [38] focussing on the eight shared disease classes. We use a train, validation, test split of (70, 10, 20). For training and validation, we filter out images with multiple pathology labels to enable a comparison with SOTA approaches like EDM-2 and EDM-2-AG.

**Metrics:** To assess the quality of image generation we use the Fréchet Inception distance (FID) that compares features extracted from a pre-trained Inception model between real and synthetic data. To assess diversity we use image retrieval score (IRS), a recently proposed method, that treats image generation as an image retrieval problem and measures how many images of the real dataset can be retrieved using synthetic samples [11]. Finally, to assess utilization, we train a downstream model for multi-class classification and report the AUCROC score on real data, selecting the best model using a validation set. Specifically, we use DenseNet-121 [18], following the approach suggested by [32]. We train models on real data $\mathcal{C}_r(x)$ and compare them to models trained on synthetic data $\mathcal{C}_s(x)$. All models are trained for 100 epochs with annealing learning rate. The best checkpoint is chosen based on the validation loss on real data (mimicing a scenario where one hospital has access to all synthetic data and one in-house validation dataset).

**Image Generation Benchmark:** For benchmarking, we use the recently established EDM [25] and its autoguidance extension EDM-AG [24], along with DiADM, a diversity-aware diffusion model designed to improve diversity over the unconditional baseline [11]. We set the learning rate to 0.0003, apply decay after 17,000 steps, and disable half-precision training due to observed inaccuracies. All models are trained for two days on four Nvidia H100 GPUs, selecting the best EDM-2 and EDM-2 AG models based on FID. For compression, we use the VAE from Stable Diffusion v2 (SDv2) without fine-tuning but compute dataset-specific latent statistics following [25]. Conditional models are trained for $83\,886$ steps, and unconditional models for $100\,663$ steps. DiADM does not use guidance (equivalent to a strength of 1.0), but we observe that adding guidance improves IRS and FID scores. We compare guidance strategies using an unconditional model [25], an earlier checkpoint of the same model [24], and a combination of both.

**Results:** First we investigate our proposed changes to the general architecture and sampling of DiADM introduced in Section 3. Our best-performing model uses a fully trained unconditional model with a guidance strength of 1.2. The results are shown in Tab. 1. While previous methods exhibit a gap of more than three percentage points in downstream AUCROC, our method reduces this gap to below one percentage point compared to real data. Our approach achieves the best performance in both image fidelity and diversity. Notably, it even surpasses an IRS value of one, indicating that conditioning on $c_s$ is effective, as the sampling exceeds the expected diversity of a perfect unconditional model.

**Table 1.** IRS and FID scores for $\mathcal{D}'$ generated from different state of the art class-conditional approaches without ensuring PSO-secure synthetic data. No augmentation or balancing technique was used.

| Name | FID ↓ | IRS$_{\infty,a}$ ↑ | Real-Snth Gap (AUCROC ↑) |
|---|---|---|---|
| EDM-2 (CVPR24) [25] | 15.0 | 0.19 | -4.49 (80.44) |
| EDM-2 AG (T/10) (Neurips24) [24] | 14.7 | 0.23 | -3.50 (81.49) |
| DiADM (CVPR25) [11] | 8.9 | 0.33 | -3.68 (81.31) |
| DiADM + SwAV (Ours) | **5.0** | **1.58** | **-0.95 (84.04)** |
| Real | — | — | 84.99 |

Now, to investigate how our PSO-secure sampling impacts model performance, we examine the results of downstream models trained on all three datasets separately and on a combination of them. The results are shown in Tab. 2. We observe that our model outperforms all others by a large margin. To statistically verify our results, we perform a ten-fold cross-validation using the training dataset. Each generative model samples one synthetic dataset, $\mathcal{D}'$, which is then split according to the ten-fold cross-validation. We ignore subject overlap for

**Table 2. Generalization to new datasets**. Data sharing (DS) means we combine privacy-preserving synthetic datasets to a large synthetic dataset.

| Test | PSO-s. | NIH | | | CXR | | | CXP | | |
|---|---|---|---|---|---|---|---|---|---|---|
| Train | | NIH | CXR | CXP | NIH | CXR | CXP | NIH | CXR | CXP |
| Real | | 85.41 | 81.78 | 79.62 | 77.07 | 82.71 | 76.98 | 74.18 | 76.23 | 79.99 |
| Rec. (SDv2) | | 85.38 | 82.47 | 81.61 | 77.82 | 82.89 | 77.85 | 74.45 | 76.21 | 79.90 |
| EDM-2 | | 80.44 | 77.89 | **77.23** | 73.85 | 80.31 | **75.81** | 67.56 | 70.74 | 73.94 |
| EDM-2 AG | | 81.49 | 73.17 | 76.24 | 74.90 | 76.12 | 74.46 | 68.21 | 70.59 | 75.02 |
| DiADM | | 81.31 | 78.52 | 74.97 | 72.88 | 80.41 | 73.12 | 67.40 | 73.97 | 73.97 |
| Ours | ✓ | **83.65** | **79.60** | 76.16 | **75.98** | **80.92** | 74.12 | **72.57** | **74.88** | **77.87** |
| Ours + DS. | ✓ | | 83.83 | | | 81.31 | | | 77.94 | |

this experiment, which results in higher scores for real data. The results are presented in Fig. 3. Importantly, we see a significant improvement in our method compared to all previously proposed methods. However, the model is still not on par with real data. We believe this may be because pseudo-conditional labels capture visual features well but do not fully represent the underlying distribution of the diseases.

To better understand why this works so well, we visualize the generated samples in Fig. 4 together with their privacy prediction and their predicate alignment. As we can see, all samples share key visual characteristics but differ in smaller details such as ribs, support devices, or heart shape. Clearly, the model does not memorize entire samples, but the visual appearance is very similar across all samples. While different memorization detec-

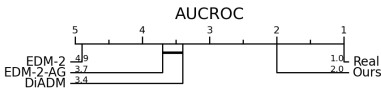

**Fig. 3.** Critical difference diagram of different generation methods [21].

tion methods might lead to different results, the privacy filter we use achieves a combined test performance of 96% AUCROC on re-identification, which is much harder than simple memorization detection. Visual inspection also gives us insight how this method ensures PSO-secure datasharing. Despite knowing all predicates about the patient (such as gender, size, existence of support devices, presence of a disease etc.) it is impossible to say which image comes from the real patient.

Given the promising results of using domain-agnostic feature encoders for pseudo-conditional generation we also experiment with using models trained on medical data. Specifically, we experiment with using $\mathcal{C}_r(x)$ as feature extractor. Suprisingly, the model does not properly learn to generate images from these features. Both IRS and FID increase by a magnitude and the downstream performance is worse than all the results presented in 1. We believe that this is

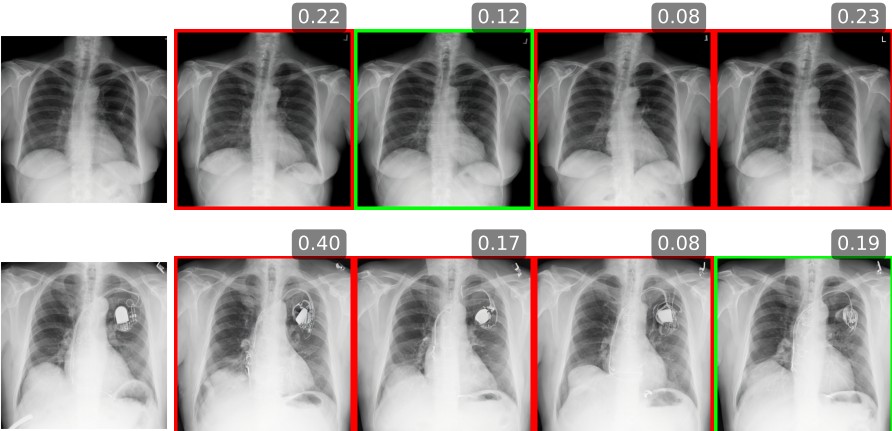

**Fig. 4.** Two random examples of four generated samples using our proposed generation method. The left-most image shows the training image. All other images have a boundary indicating the prediction of $\mathcal{P}(x, x')$, with red indicating privacy issues. The scores indicate the agreement of real and PSO-secure image according to 1.

because the pseudo-conditional features extracted by this domain-specific model are visually not meaningful enough for the diffusion model.

**Limitations:** Using our framework could lead to synthetic datasets that are as good as real datasets. However, we did not demonstrate how a generative model can outperform real data, limiting the impact of our approach in typical applications of generative models, such as data augmentation. Additionally, the results highly depend on the privacy filters. Nevertheless, we have successfully shown that our filtering approach can generate images that are sufficiently different to achieve generalization according to a privacy filter. Sampling time when using filtering increases by a factor of $b$. We believe this is acceptable if it results in privacy-preserving datasets.

## 5   Conclusion

We propose a method for generating synthetic, privacy-preserving datasets that retain competitive downstream performance on image classification tasks compared to real data. Our approach outperforms state-of-the-art class-conditional methods across multiple metrics and datasets. To ensure privacy, we formalize the concept of "singling-out", *i.e.*, the risk of identifying individuals based solely on predicates, and explicitly prevent it, paving the way for a new paradigm in secure data sharing.

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
