# OpenReview forum: "Enabling PSO-Secure Synthetic Data Sharing Using Diversity-Aware Diffusion Models"
_MICCAI.org/2025/Workshop/BRIDGE — BRIDGE 2025 Poster_

### Official Review · Reviewer_9bQa · 2025-07-24
**Review Comments**

**Rating:** 7
**Confidence:** 4

**Review:**

I appreciate your effort to balance privacy and utility with synthetic chest X-rays. Overall, the paper is on the right track, but it needs more depth in several areas.
1.	Your idea of using predicate singling-out (P(x, x′)=0) to meet GDPR is clever, but you haven’t convinced me it’s better than standard methods like membership-inference tests or differential privacy. You should add at least a paragraph explaining why predicate filtering is more reliable or efficient, and then run an experiment against a stronger attacker, like a more powerful re-identification (re-ID) network or an adversarial scheme, to show that your filter still blocks real identities.
2.	The implementation details are vague. You need to list the exact diffusion model architecture, the SwAV backbone, the re-ID network you used, and all hyperparameters including learning rates, batch sizes, and β-schedule for the diffusion. Also, you need to state the random seeds and include [mean ± std] over at least three runs for FID, IRS, and AUCROC. Right now, “reduce classifier guidance by 0.1” sounds chosen at random; you must explain how you picked 0.1 and show what happens if you use other values instead.
3.	You only test on an eight-way pathology classifier. That’s a good start, but it doesn’t show whether your synthetic data works for other tasks. Please add at least one more downstream experiment, like lesion segmentation or object detection on the same dataset, to prove your point holds beyond classification. You may need to break down the AUC by disease class and check whether synthetic data under- or over-represent rare conditions or certain patient groups.
4.	I’d like to see a short section on fairness. Does your synthetic data introduce bias? You could compare class distributions and classification errors across age, sex, or acquisition device. You can add a table showing AUC by subgroup.
5.	The paper’s presentation needs polishing. Define all symbols (Cr(x), Cs(x), P(x, x′)) in one place. Move your notation to its subsection. In Figures 1 and 2, add the dataset size and hyperparameter values to the captions. In Tables 1 and 2, include confidence intervals or p-values so I know if the differences are significant.
6.	You could mention that predicate filtering might not cover all privacy attacks, that more diverse patient demographics need testing, and that the method has only been tried on X-rays so far.

---

### Official Review · Reviewer_tQoM · 2025-07-25
**Reviewer Comments**

**Rating:** 7
**Confidence:** 4

**Review:**

## Summary
This paper presents a framework for generating PSO-secure synthetic medical imaging datasets using diversity-aware diffusion models that achieve performance within one percentage point of real data while ensuring GDPR compliance. The method employs SwAV feature encoders for pseudo-conditional generation and privacy filters to prevent patient re-identification, outperforming existing approaches on multiple chest X-ray datasets.

## Strengths
List 2–3 strong points in the paper (e.g., novelty, clarity, empirical rigor, real-world relevance).
* The authors address critical medical data sharing needs by explicitly incorporating GDPR compliance through the PSO-security framework.
* The paper demonstrates a reduction in real-synthetic performance gap across multiple datasets with comprehensive evaluation (both image quality and task-based assessment).
* This work combines diversity-aware diffusion models with privacy filtering to simultaneously optimize data utility and legal privacy protection through an innovative selection approach.

## Limitations or Areas for Improvement
Are there any gaps, unclear sections, missing experiments, or methodological issues?
* Their PSO-security claims could be stronger with evaluation against multiple privacy attack methods and formal privacy guarantees beyond re-identification.
* The paper lacks explanation for why SwAV features work better than domain-specific medical features for pseudo-conditional generation.
* The privacy function P(x, x') in Equation 1 is not clearly explained. The paper states P(x, x') = 0 ensures "knowing predicates cs does not imply knowledge of the identity" but provides insufficient detail about how this binary privacy decision is computed or what specific thresholds are used.

## Relevance to BRIDGE Workshop Topics
How well does the paper align with the themes of real-world evaluation, robustness, generalizability, or interpretability?
* The paper excellently addresses practical GDPR compliance needs and tests on actual clinical datasets.
* The evaluation is restricted to chest X-ray datasets with no demonstration of broader applicability.

---

### Official Review · Reviewer_wSbY · 2025-07-25
**Reviewer's Comments**

**Rating:** 7
**Confidence:** 3

**Review:**

### 1. Summary of the Paper
This paper presents a way to generate synthetic medical images that protect patient privacy while still being useful for research and clinical applications. The approach uses diversity-aware diffusion models with predicate-based conditioning methods and includes a filtering step to prevent any patient identification. The method meets GDPR privacy requirements through what the authors call "Predicate Singling-Out security."

### 2. Strengths
* The method tackles the important issue of legal data sharing in healthcare by connecting technical methods with GDPR compliance.
* The paper does a good job explaining the connection between legal definitions of identifiability and the goals of the generative framework.

### 3. Limitations or Areas for Improvement
* Adding more details on how P(x, x′) is computed would be helpful for the reader.
* The paper could benefit from reporting precision and recall for generative models, and from including label distribution comparisons to assess whether class balance is preserved in the synthetic data—especially relevant in clinical tasks with skewed disease prevalence.

### 4. Relevance to BRIDGE Workshop Topics
* The method is directly motivated by legal compliance and tested on multiple real-world datasets, making it highly aligned with BRIDGE themes.

---

### Decision · Program_Chairs · 2025-07-25

**Decision:**

Accept (Poster)

**Comment:**

Dear Authors,

Congratulations!

We are pleased to inform you that your paper has been accepted for the BRIDGE Workshop. Your submission was reviewed by three scientists from regulatory, academic, and industry backgrounds to provide diverse perspectives. Please see the reviewers' comments below.

Requirements for your final camera-ready submission (due July 30):
* Incorporate reviewer comments and suggestions where appropriate throughout your paper. At minimum, add a discussion section that acknowledges and responds to the key points raised by reviewers
* Ensure your final draft follows MICCAI conference and Springer formatting guidelines
* Submit your camera-ready source files and any supplementary materials


We look forward to your presentation and the engaging discussions it will generate at the workshop!

Best regards,
BRIDGE Workshop Organizers